# Dose-Dependent Beneficial Effects of Tryptophan and Its Derived Metabolites on *Akkermansia* In Vitro: A Preliminary Prospective Study

**DOI:** 10.3390/microorganisms9071511

**Published:** 2021-07-14

**Authors:** Jia Yin, Yujie Song, Yaozhong Hu, Yuanyifei Wang, Bowei Zhang, Jin Wang, Xuemeng Ji, Shuo Wang

**Affiliations:** Tianjin Key Laboratory of Food Science and Health, School of Medicine, Nankai University, Tianjin 300071, China; spyinjia@163.com (J.Y.); yjsongqcyy@163.com (Y.S.); yzhu@nankai.edu.cn (Y.H.); wangyyf163@163.com (Y.W.); bwzhang@nankai.edu.cn (B.Z.); wangjin@nankai.edu.cn (J.W.); jixuemeng@nankai.edu.cn (X.J.)

**Keywords:** *Akkermansia*, tryptophan metabolites, indole, indole derivates

## Abstract

*Akkermansia muciniphila*, a potential probiotic, has been proven to lessen the effects of several diseases. As established, the relative abundance of *Akkermansia* is positively correlated with tryptophan metabolism. However, the reciprocal interaction between tryptophan and *Akkemansia* is still unclear. Herein, for the first time, the possible effects of tryptophan and its derived metabolites on *A. muciniphila* were preliminarily investigated, including growth, physiological function, and metabolism. Obtained results suggested that 0.4 g/L of tryptophan treatment could significantly promote the growth of *A. muciniphila*. Notably, when grown in BHI with 0.8 g/L of tryptophan, the hydrophobicity and adhesion of *A. muciniphila* were significantly improved, potentially due to the increase in the rate of cell division. Furthermore, *A. muciniphila* metabolized tryptophan to indole, indole-3-acetic acid, indole-3-carboxaldehyde, and indole-3-lactic acid. Indoles produced by gut microbiota could significantly promote the growth of *A. muciniphila*. These results could provide a valuable reference for future research on the relationship between tryptophan metabolism and *A. muciniphila*.

## 1. Introduction

Trillions of microorganisms colonize the human gastrointestinal (GI) tract, where the numerous metabolites produced can have marked effects on host physiology. The Gram-negative anaerobe, *Akkermansia muciniphila* (*A. muciniphila*), is a member of the phylum Verrcomicrobia. It can utilize mucin as a sole source of carbon and nitrogen and has been identified as part of the human gut microbiome [1]. Several studies have demonstrated that *A. muciniphila* is able to ameliorate the effects of numerous diseases, such as colitis, obesity, diabetes, and aging [2,3,4,5]. The probiotic effects of *A. muciniphila* are always accompanied by changes in tryptophan metabolism. Similarly, a previous study suggests that *A. muciniphila* plays a vital role in tryptophan metabolism [6]. In nonalcoholic fatty liver disease (NAFLD) mice, the relative abundance of *A. muciniphila* was decreased, as was the level of tryptophan metabolites, such as indole, indole-3-acetic acid, and indole-3-propionic acid, which act as ligands for aryl hydrocarbon receptor (AHR) to protect intestinal mucosal homeostasis [7]. Disturbances to gut microbiota always emerge in children with autism spectrum disorder (ASD). It has been reported that the relative abundance of beneficial bacteria, such as *Clostridium XlVa* and *A. muciniphila*, were decreased in children with ASD. Meanwhile, tryptophan metabolism was significantly downregulated in comparison to that of healthy children [8]. In addition, the correlation analysis showed that *A. muciniphila* was closely correlated to tryptophan biosynthesis in depressive rats [9]. Therefore, based on the above, we hypothesized that the positive effects of *A. muciniphila* may be correlated with tryptophan metabolism.

L-tryptophan is one of the essential amino acids, which cannot be synthesized by humans, but can only be supplemented through dietary intake. Tryptophan can be catabolized by the kynurenine pathway to kynurenine (KYN), kynurenic acid, and quinolinic acid in the host. However, a small fraction of tryptophan can reach the colon, which is then metabolized to indole and indole derivatives by gut microbiota [10]. This is significant, as current studies have shown that indole and its derivates produced by intestinal bacteria have many positive effects on human health. Moreover, indoles bind with AHR to increase the level of IL-22, which alleviates intestinal mucosal barrier dysfunction caused by inflammation [11]. Additionally, indoles could activate AHR to upregulate the production of IL-10, which is beneficial for promoting goblet cell differentiation during aging [12]. Notably, different bacterial species have been proven to be able to catabolize tryptophan into different indoles [13]. However, there is no research into the tryptophan metabolites produced by *A. muciniphila*.

In our previous study, we found that tryptophan supplementation through the diet could enhance the relative abundance of *A. muciniphila* during aging [14]. Thus, we speculated that supplementation of tryptophan might be beneficial to the growth of *A. muciniphila*. However, the relationship between tryptophan metabolism and *Akkermansia* has not been investigated. Therefore, in this study, we aimed to elucidate the interaction between tryptophan and *Akkermansia*. Our results revealed that supplementation of tryptophan increased the hydrophobicity and adhesion of *A. muciniphila* BAA-835^T^. Meanwhile, *A. muciniphila* could metabolize tryptophan to indole, IAA, Icld, and ILA, which potentially contributes to the probiotic functions of *A. muciniphila*. Furthermore, the growth of *A. muciniphila* could be promoted by specific indoles produced by other members of the gut microbiome.

## 2. Materials and Methods

### 2.1. Chemicals and Materials

*Akkermansia muciniphila* BAA-835^T^ was purchased from Guangdong Microbial Culture Collection Center (GDMCC) (Guangzhou, China). Indole-3-propionic acid (IPA), indole-3-carboxaldehyde (Icld), indole-3-acetaldehyde (Iald), indole-3-acetamide (IAM), 3-methylindole (3ML), tryptamine (TA), kynurenine (KYN), and xylene were purchased from Aladdin (Shanghai, China). Indole and indole-3-acrylicacid (IA) were obtained from Macklin (Shanghai, China). Tryptophan (Trp), indole-3-acetic acid (IAA), and 5-hydroxytryptamine (5-HT) were obtained from Shanghai yuanye Bio-Technology Co., Ltd. (Shanghai, China). Indole-3-lactic acid (ILA) was purchased from Rhawn (Shanghai, China). 5-(and 6)-carboxyfluorescein diacetate succinimidyl ester (CFDA-SE, CFSE) was purchased from Invitrogen (Carlsbad, CA, USA).

### 2.2. Bacterial Strains and Culture Conditions

*Akkermansia muciniphila* BAA-835^T^ was cultured in BHI-medium (Oxoid, Basingstoke, UK), supplemented with 0.05% L-cysteine at 37 °C for 2 days in an anaerobic incubator (gas mix of 5% CO_2_, 5% H_2_, and 90% N_2_). Tryptophan, indole, and indole derivatives were added to the medium with different concentrations. The optical density (OD_600_) of cultures were monitored after appropriate incubation times. Triplicate cultures were performed in all experiments at 37 °C under anaerobic conditions.

### 2.3. Auto-Aggregation and Hydrophobicity Analysis

*A. muciniphila* BAA-835^T^ was cultured at 37 °C for 24 h in an anaerobic incubator in BHI, containing 0.2 g/L of tryptophan (Appendix A). In order to investigate the effect of tryptophan on the growth of *A. muciniphila* BAA-835^T^, the BHI medium was supplemented with 0, 0.2, 0.4, and 0.6 g/L of tryptophan to the final concentration of 0.2 (Ctrl), 0.4, 0.6, and 0.8 g/L of tryptophan, respectively. Bacteria were harvested by centrifugation at 6000× *g* for 5 min at 4 °C. The harvested cells were washed twice and resuspended in PBS (pH = 7.4). Bacterial suspensions were divided into 4 tubes and incubated in the anaerobic environment for 8 h. The OD_600_ was determined by spectrophotometer every 2 h. The auto-aggregation was calculated as follows: $\left[\left(1-{\mathrm{OD}}_{\mathrm{upper}\;\mathrm{suspension}}\right)\right]/{\mathrm{OD}}_{\mathrm{initial}}\times 100\%$.

For the hydrophobicity test, l mL of cell suspension was mixed with an equal volume of xylene and vortexed for 5 min. The OD_600_ of the aqueous phase was measured after the solution fully stratified. The hydrophobicity of *A. muciniphila* BAA-835^T^ was calculated as follows: $\left[\left(1-{\mathrm{OD}}_{\mathrm{aqueous}\;\mathrm{phase}}\right)\right]/{\mathrm{OD}}_{\mathrm{initial}}\times 100\%$.

### 2.4. Scanning Electron Microscopy (SEM)

The surface morphology of *A. muciniphila* BAA-835^T^ after treatment with tryptophan was determined by scanning electron microscopy (SEM). *A. muciniphila* BAA-835^T^ was cultured for 24 h in BHI supplemented with different concentrations of tryptophan, where cells were collected by centrifugation at 6000× *g* for 5 min at 4 °C, then washed twice with sterile PBS. The bacterial cells were fixed with 2.5% (*v*/*v*) glutaraldehyde at 4 °C for 24 h. After that, cells were washed in sterile PBS and dehydrated in a graded ethanol series for 15 min in each concentration. Cells were then freeze-dried and sputter-coated with gold for SEM observation.

### 2.5. Epithelial Cell Lines

The HT-29 cell line was cultured in Dulbecco’s modified Eagle’s medium (DMEM) containing 4.5 g/L of glucose (Thermo Fisher scientific, Waltham, MA, USA), supplemented with 10% fetal bovine serum (Thermo Fisher scientific, Waltham, MA, USA) and 1% penicillin/streptomycin (Thermo Fisher scientific, Waltham, MA, USA) at 37 °C in a humidified 5% CO_2_ incubator. The cells were sub-cultured every 2–3 days.

### 2.6. Bacterial Adhesion to HT-29 Cells

For the adhesion assay, cells were seeded in a 24-well plate at a density of 1 × 10^5^ per well and grown to ~80% confluence. *A. muciniphila* BAA-835^T^ was cultured at 37 °C for 24 h in an anaerobic incubator in BHI supplemented with different concentrations of tryptophan. The bacterial cells were washed twice with PBS, and then resuspended in DMEM without antibiotics at a concentration of ~10^8^ CFU/mL. HT-29 cells were washed, and aliquots of bacterial suspension were added to the wells. The plate was incubated at 37 °C with 5% CO_2_ for 1 h. After that, the wells were washed three times with PBS, removing the unattached bacteria. The cells with adherent bacteria were released using 0.25% trypsin (Thermo Fisher scientific, Waltham, MA, USA). Bacterial counts were measured by plating on BHI plates by 10-fold serial dilutions. Adhesion ability was calculated as follows: the number of adherent bacteria/the number of bacteria inoculated × 100%. Triplicate experiments were performed.

### 2.7. UPLC/Q-TRAP MS Method for Tryptophan Metabolomics of A. muciniphila

*A. muciniphila* BAA-835^T^ was grown at 37 °C for 24 h in an anaerobic environment, the supernatant was collected and separated into tubes, then 100 μL of the supernatant was mixed with 300 μL of 50% methanol (containing 0.1% formic acid), followed by vortexing the samples and then centrifuging. The supernatant was then filtered by 0.22 μm membrane. The concentrations of tryptophan metabolites were separated and analyzed using the method described in the previous study [15]. Operating parameters in MRM-mode are shown in Appendix A.

### 2.8. Statistical Analysis

The data were analyzed using GraphPad Prism and expressed as the mean ± standard deviation (SD). Statistical differences between two groups were analyzed by an unpaired Student’s *t*-test. Results with more than two groups were compared using a one-way ANOVA test, followed by a Dunnett’s post hoc test. *p* < 0.05 was considered as statistically significant.

## 3. Results

### 3.1. The Effect of Tryptophan on the Growth of A. muciniphila

Numerous studies have shown that the relative abundance of *A. muciniphila* is correlated with tryptophan metabolism. To explore whether tryptophan could promote the growth of *A. muciniphila* BAA-835^T^, we added 0.2, 0.4, and 0.6 g/L of tryptophan into BHI medium to observe the growth of *A. muciniphila* BAA-835^T^. The growth rate constants of *A. muciniphila* BAA-835^T^ with different tryptophan concentrations were measured by a spectrophotometer (λ = 600 nm). As depicted in Figure 1A, there was no significant difference of OD_600_ between the four groups in the stationary phase. However, in comparison with the Ctrl group, the OD_600_ of the 0.4 g/L Trp group was significantly higher than that of the Ctrl group at 20 and 24 h (*p* < 0.05) (Figure 1B). Meanwhile, in contrast to the Ctrl group, the colony forming units (CFU) at 24 h was significantly improved in the 0.4 g/L Trp group (Figure 1C). These results indicate that a moderate supplementation of tryptophan could significantly advance the logarithmic growth phase, which benefits the growth and reproduction of *A. muciniphila*.

### 3.2. The Effect of Tryptophan on the Auto-Aggregation, Hydrophobicity, and the Surface Morphology of A. muciniphila

In order to investigate the influence of tryptophan on the colonization of *A. muciniphila* BAA-835^T^ in the gut, auto-aggregation and hydrophobicity experiments were performed. As illustrated in Figure 2A, the auto-aggregation ability of *A. muciniphila* BAA-835^T^ cultured with different doses of tryptophan had no significant difference as compared with the Ctrl group (*p* > 0.05).

Cell surface hydrophobicity is always influenced by a complex interplay between positive/negative charges and hydrophobic/hydrophilic structures on the surface of microbes [16]. Thus, we tested the hydrophobicity of *A. muciniphila* BAA-835^T^ with different concentrations of tryptophan. In comparison with the Ctrl group, the hydrophobicity of *A. muciniphila* BAA-835^T^ cultured in the BHI medium containing 0.8 g/L tryptophan was significantly increased to 70.5892% ± 1.5533%, from 60.6025% ± 2.8638% (*p* < 0.05) (Figure 2B). These results indicate that the supplement of tryptophan could improve the adhesion ability of *A. muciniphila* BAA-835^T^, while greater adhesion ability is conducive to the colonization of *A. muciniphila* BAA-835^T^ in the gut.

To investigate the changes to surface morphology of *A. muciniphila* BAA-835^T^ when grown with tryptophan, the surface morphology of *A. muciniphila* BAA-835^T^ cultured with different concentrations of tryptophan was determined by scanning electron microscopy (SEM). As shown in Figure 2C, after treatment with 0.4 g/L of tryptophan, *A. muciniphila* BAA-835^T^ had a bigger size as compared with the Ctrl group. However, treatment with 0.6 and 0.8 g/L of tryptophan led to smaller diameters of cells than those grown in the control medium. This phenomenon determined that 0.4 g/L of tryptophan might improve the rate of cell division of *Akkemansia muciniphila*, whereas 0.6 and 0.8 g/L potentially increased the adhesion ability of *Akkermansia miniphila*.

### 3.3. The Effect of Tryptophan on the Adhesion Ability of A. muciniphila to HT-29 Cells

Due to the increase in hydrophobicity of *A. muciniphila* BAA-835^T^ after supplementation of tryptophan, we concluded that tryptophan could increase the adhesion ability of *A. muciniphila* BAA-835^T^. To understand whether the supplementation of tryptophan could influence the adhesion of *A. muciniphila* BAA-835^T^ to intestinal epithelial cells, we conducted adhesion assays using the HT-29 cell line. As depicted in Figure 3A, the adhesion ratio of *A. muciniphila* BAA-835^T^ in the Ctrl group was 0.5153% ± 0.1192%, and the adhesion ratios of cells supplemented with tryptophan to HT-29 cells were 0.5438% ± 0.0599%, 0.7349% ± 0.0522%, and 1.2789% ± 0.3522%, respectively. These data indicated that 0.8 g/L of tryptophan significantly increased the adhesion ability of *A. muciniphila* BAA-835^T^ compared with the Ctrl group (0.5153% ± 0.1192%) (*p* < 0.05).

Additionally, bacterial cells cultured with various concentrations of tryptophan were labelled by CFDA-SE and were loaded into HT-29 cells and incubated at 37 °C for 1–2 h to observe the difference in adhesion ability with a fluorescent inverted microscope. We observed that the number of *A. muciniphila* BAA-835^T^ attached to HT-29 cells were more than that of the Ctrl group after supplementation with 0.6 and 0.8 g/L of tryptophan (Figure 3B). These data provide evidence that the supplementation of tryptophan could increase the adhesion ability of *A. muciniphila* BAA-835^T^.

### 3.4. The Tryptophan Metabolites of A. muciniphila

To explore the tryptophan metabolites of *A. muciniphila* BAA-835^T^, the HPLC/Q-TRAP MS method was used to analyze the contents of 9 common tryptophan metabolites in the medium after 24 h of incubation (AKK group). Meanwhile, the BHI medium inoculated with *A. muciniphila* BAA-835^T^ but not incubated was used as the control group. The supernatants of the control and the AKK group were collected, and the metabolites were extracted and analyzed. As shown in Figure 4A, compared with the Ctrl group, the levels of indole, IAA, and Icld were significantly increased in the AKK group (*p* < 0.05). Meanwhile, ILA was not detected in the Ctrl group, but existed in the AKK group. It has been well-established that ILA has great effects in improving gastrointestinal tract health, while this study has shown that tryptophan could be metabolized into ILA by *A. muciniphila* BAA-835^T^. However, the level of IPA was significantly decreased in the AKK group compared with the Ctrl group (*p* < 0.05). Moreover, no significant difference in the levels of Trp, KYN, 5-HT, IA, and IAM between the two groups (*p* > 0.05) was observed (Figure 4B), indicating that *A. muciniphila* BAA-835^T^ could metabolize tryptophan to produce indole, IAA, Icld, and ILA, which have beneficial effects on health. 

### 3.5. The Effects of Tryptophan Metabolites Produced by Akkermansia on the Growth of A. muciniphila BAA-835^T^

To determine the effects of tryptophan metabolites produced by *Akkermansia* on the growth of *A. muciniphila* BAA-835^T^, we added several different concentrations of indole, IAA, Icld, and ILA into BHI medium, respectively. The OD_600_ of *A. muciniphila* BAA-835^T^ was measured after samples were anaerobically cultured for 24 h. As shown in Figure 5, the supplementation of indole and IAA significantly promoted the growth of *A. muciniphila* BAA-835^T^ (*p* < 0.05). However, the supplementation of Icld and ILA had no significant effect on the growth of *A. muciniphila* BAA-835^T^ (*p* > 0.05). Our results showed that indole and IAA, which can be produced by *Akkermansia*, could promote the growth of *A. muciniphila* BAA-835^T^.

### 3.6. The Effects of Tryptophan Metabolites Produced by Other Gut Microbes on the Growth of A. muciniphila BAA-835^T^

Tryptophan can be catabolized to varieties of indole and indole derivates by gut microbiota. In order to investigate the effects of indole metabolites of other gut microbes on the growth of *A. muciniphila* BAA-835^T^, we supplemented IPA, Iald, IA, IAM, TA, and 3-ML into BHI medium respectively, and the OD_600_ of *A. muciniphila* BAA-835^T^ was measured after samples were anaerobically cultured for 24 h at 37 °C. As shown in Figure 6A, the supplementation of IPA, IA, IAM, and TA could promote the growth of *A. muciniphila* BAA-835^T^, whereas this effect was not observed after adding Iald or 3ML. Furthermore, 5-HT and KYN, which were produced by the host, were not observed to promote the growth of *A. muciniphila* BAA-835^T^ (Figure 6B). These data suggested that tryptophan metabolites produced by other bacteria in the gut were conducive to the growth of *Akkermansia muciniphila*.

## 4. Discussion

*Akkermansia muciniphila* has been considered as a potential probiotic, and possesses significant medical effects on human health [3,4,17,18,19]. The relative abundance of *A. muciniphila* is related with tryptophan metabolism. However, the relationship between *A. muciniphila* and tryptophan metabolism remains unclear. In the present study, we investigated the effects of tryptophan and its metabolites on *A. muciniphila*.

Our previous study showed that the 0.4% tryptophan diet raised the relative abundance of *Akkermansia* in aging mice, but the 0.8% tryptophan diet had no significant effect [14]. Consistent with this point, we observed that the growth of *A. muciniphila* BAA-835^T^ was improved in log phase with 0.4 g/L of tryptophan, but not in the 0.6 or 0.8 g/L groups in this study. Thus, we considered that 0.4 g/L of tryptophan could promote the growth of *A. muciniphila* in vitro.

Auto-aggregation and cell surface hydrophobicity are usually used to analyze the surface properties of bacteria [20]. Hydrophobic amino acids, especially aromatic amino acids, can protect the surface protein of bacteria against an attack from external environmental stress by building a hydrophobic area [21]. A previous study has proven that aromatic amino acids were more abundant when *B. longum* BBMN68 was under bile exposure. Meanwhile, the cell surface hydrophobicity of the strain was significantly increased compared with the control [22]. In this study, tryptophan is a kind of aromatic amino acid, and supplementation of tryptophan could significantly improve the hydrophobicity ability of *A. muciniphila* BAA-835^T^. Thus, we speculated that tryptophan could enhance the cell surface hydrophobicity of *A. muciniphila*, potentially through regulating the related metabolic pathway.

Furthermore, cells in the 0.4 g/L tryptophan group had a larger diameter compared with the Ctrl group, which conformed with the methods proposed by previous studies [23,24], and the faster a bacterial cell grows, the bigger the cell morphology. We also observed that the size of *A. muciniphila* BAA-835^T^ in both 0.6 and 0.8 g/L tryptophan groups became smaller than Ctrl after tryptophan intervention. In view of these results, we investigated whether 0.4 g/L of tryptophan could accelerate the rate of cell division, which leads to a bigger cell morphology, whereas *A. muciniphila* BAA-835^T^, cultured with 0.6 and 0.8 g/L of tryptophan, have a higher hydrophobicity ability and a smaller size, which potentially improve the adhesion ability of *A. muciniphila* BAA-835^T^. To sum up, supplementation with different doses of tryptophan has different effects on cell size, which are potentially closely related to growth, hydrophobicity ability, and adhesion ability of *A. muciniphila* BAA-835^T^.

Adhesion to intestinal cells and colonization of the gastrointestinal tract are necessary for probiotics to exert positive effects on the host. As a potential probiotic, the colonization of *A. muciniphila* is fundamental to good health. Moreover, improvement to the adhesion ability of *Akkemansia muciniphila* to intestinal cells might be efficient in enhancing the positive effects on human health. Notably, a previous study has demonstrated that *Akkermansia muciniphila* could strongly adhere to enterocytes, and could significantly improve enterocyte monolayer integrity [25]. Thus, improvement to the adhesion ability of *Akkermansia muciniphila* could strengthen the integrity of the epithelial cell layer. This study suggested that the supplementation of tryptophan could significantly increase the adhesion ability of *A. muciniphila* BAA-835^T^ to intestinal cells. Therefore, we speculated that tryptophan could promote the adhesion of *Akkermansia muciniphila* to intestinal cells to protect the intestinal barrier integrity. Possible mechanisms will be investigated in a future study. More significantly, improvement to cell surface hydrophobicity has been proven to be beneficial for boosting the adhesion ability of bacteria [26]. Thus, it was evident why the trend in adhesion ability positively correlated with that of surface hydrophobicity.

It has been reported that supplementation of *A. muciniphila*^sub^, an *Akkermansia muciniphila* subtype, could improve the level of 5-hydroxytryptamine in serum, which is significantly decreased by a high-fat diet [27]. 5-hydroxytryptamine (5-HT) is an important neurotransmitter produced by tryptophan metabolism, and it can upregulate the expression of brain-derived neurotrophic factor (BDNF), which is beneficial for neurogenesis [28]. Moreover, 5-HT has a positive effect on immunity and reduces inflammation caused by mucosal infections to maintain intestinal homeostasis [10]. More importantly, dietary tryptophan restriction could decrease the relative abundance of *Akkermansia* spp. in *Ercc1*^-/Δ7^ mice. *Akkermansia* spp. possesses the genes coding for tryptophanase, which can catalyze the metabolism of tryptophan to indole [29]. Based on the above, we speculated that tryptophan metabolism was involved in the growth of *A. muciniphila*.

Tryptophan can be metabolized by intestinal bacteria to indole and its derivates, where Roager et al. identified the gut bacterial species reported to produce tryptophan catabolites [13]. However, no relevant literature regarding the products of tryptophan metabolized by *A. muciniphila* has been published. Our work determined that tryptophan could be metabolized to indole, IAA, Icld, and ILA by *A. muciniphila* BAA-835^T^. Indole has been proven to extend the lifespan via interaction with AHR and maintain intestinal barrier integrity [30,31]. Indole has multiple special functions in microbiology as a well-known intercellular signaling molecule. Indole could increase the formation of biofilm through the quorum sensing pathway [32], and our study found that supplementation of the supernatant of *A. muciniphila* BAA-835^T^ fermented broth to fresh BHI medium could significantly promote the growth of *A. muciniphila* BAA-835^T^ (Appendix A). This phenomenon indicated that indole might be an important signaling molecule in growth and adhesion of *A. muciniphila*. Furthermore, indole could decrease the biofilm and virulence of pathogenic bacteria, such as *Listeria monocytogenes* [33]. IAA can be produced by *Bifidobacterium*, which could lessen inflammatory responses in the liver caused by a high-fat diet, and these effects were dependent on the AHR pathway [34]. Meanwhile, IAA could upregulate the heme oxygenase-1 pathway to inhibit the inflammation and oxidative stress induced by LPS in RAW264.7, but these effects are independent of AHR [35]. *Lactobacillus* could metabolize tryptophan to Icld and ILA. Icld could promote the production of IL-22 via interaction with AHR, inducing the expression of IL-18 to inhibit Candida infection [36]. It has been reported that ILA produced by *Bifidobacterium* could regulate innate immunity and cell development via STAT1 pathways in immature enterocytes [37]. These small molecules might be key in explaining the interaction between tryptophan metabolism and *Akkermansia*.

Furthermore, it should be noted that the level of IPA was significantly decreased in the AKK group compared with the control group. IPA is always produced by *Clostridium* and *Peptostreptococcus*, which have been proven to have positive health effects. IPA could maintain the integrity of the intestinal barrier via pregnant X receptor and toll-like receptor 4 [38]. Meanwhile, Li et al. observed that 1-deoxynojirimycin treatment significantly increased the level of IPA in the feces of mice, and simultaneously enriched *Akkermansia* [39]. These results revealed that *A. muciniphila* might be able to utilize the tryptophan metabolites, which are produced by other commensal intestinal flora.

Importantly, we found that IPA, IA, IAM, and TA could promote the growth of *A. muciniphila* BAA-835^T^, and these results were consistent with the bioinformatics analysis which predicted that *Akkermansia muciniphila* contributes to the degradation of tryptamine and the production of 4,6-dihydroxyquinoline [6]. These findings demonstrate that *A. muciniphila* BAA-835^T^ could utilize the tryptophan metabolites which are produced by intestinal microorganisms, but not ones produced by the host.

In short, the current results suggested that supplementation of tryptophan could promote the growth and colonization of *A. muciniphila* BAA-835^T^ in the gut, with the possible mechanisms summarized in two points. On the one hand, *A. muciniphila* BAA-835^T^ could directly utilize tryptophan to improve its proliferation and adhesion ability. On the other hand, *A. muciniphila* BAA-835^T^ could also use tryptophan metabolites produced by other intestinal microorganisms to modify their own properties. In view of these interesting results, further research will focus on the interactions of tryptophan and *A. muciniphila* in vivo.

## 5. Conclusions

In summary, this preliminary study demonstrates the relationship between tryptophan and *Akkermansia muciniphila*. The supplementation of tryptophan could improve the growth, hydrophobicity, and adhesion ability of *A. muciniphila* BAA-835^T^, potentially through effects on cell division. *A. muciniphila* BAA-835^T^ could metabolize tryptophan to indole, IAA, Icld, and ILA. Meanwhile, the growth of *A. muciniphila* BAA-835^T^ could be promoted through tryptophan metabolites produced by other bacteria in the intestinal microbiome. This study determined the relationship between tryptophan metabolism and *A. muciniphila*, and showed that the supplementation of dietary tryptophan could potentially promote positive effects of *A. muciniphila* in the human intestines, but the mechanisms remain unclear. The enzymes involved in tryptophan metabolism in *A. muciniphila* also warrant future investigation. These results demonstrate that supplementation of tryptophan/tryptophan metabolites can promote the growth and adhesion ability of *A. muciniphila*. This work provides the research basis for better utilization of the probiotic effects of *A. muciniphila* in human health.

## Figures and Tables

**Figure 1 microorganisms-09-01511-f001:**
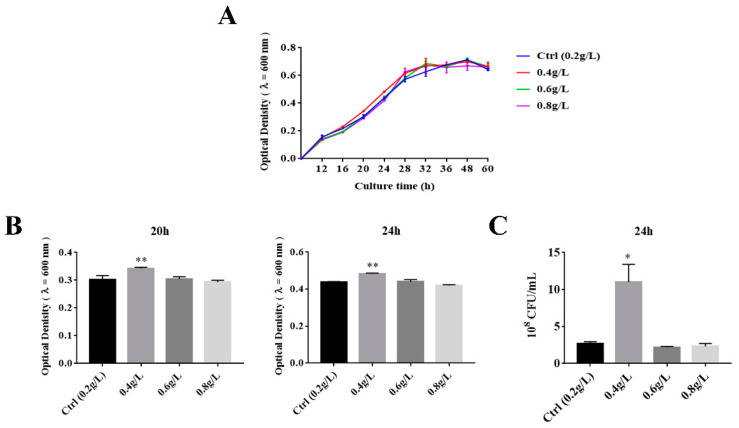
Effects of 0.2, 0.4, 0.6, and 0.8 g/L of tryptophan on growth of *A. muciniphila* BAA-835T. (**A**) The growth curve of *A. muciniphila* BAA-835T grew with different concentrations of tryptophan. (**B**) Optical density of *A. muciniphila* BAA-835T at 20 and 24 h. (**C**) CFU/mL of *A. muciniphila* BAA-835T at 24 h. One-way ANOVA with a post hoc Dunnett test was used to determine statistical differences, compared with the Ctrl group. * *p* < 0.05, ** *p* < 0.01.

**Figure 2 microorganisms-09-01511-f002:**
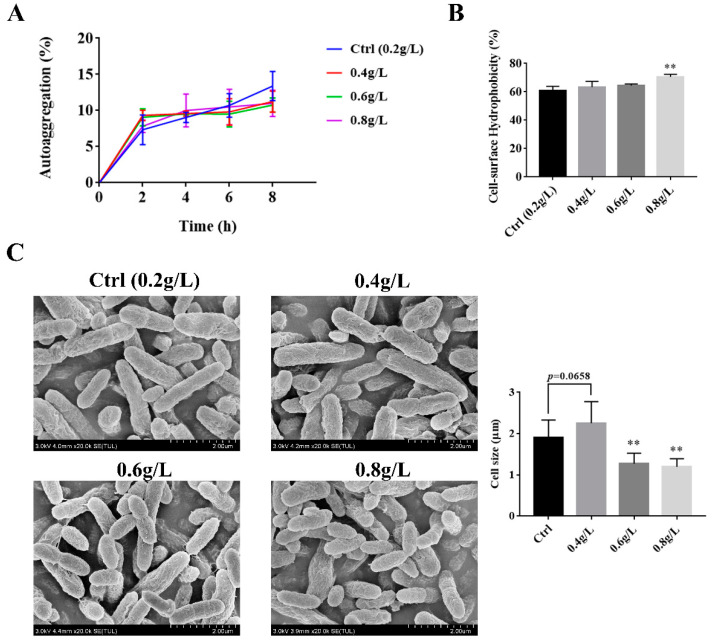
Effects of 0.2, 0.4, 0.6, and 0.8 g/L of tryptophan on auto-aggregation, hydrophobicity, and adhesion of *A. muciniphila* BAA-835^T^. (**A**) The auto-aggregation (%) of *A. muciniphila* BAA-835^T^ every 2 h. (**B**) The cell-surface hydrophobicity (%) of *A. muciniphila* BAA-835^T^. (**C**) Scanning electron microscope (SEM) of *A. muciniphila* BAA-835^T^. One-way ANOVA with a post hoc Dunnett test was used to determine statistical differences, compared with the Ctrl group. ** *p* < 0.01.

**Figure 3 microorganisms-09-01511-f003:**
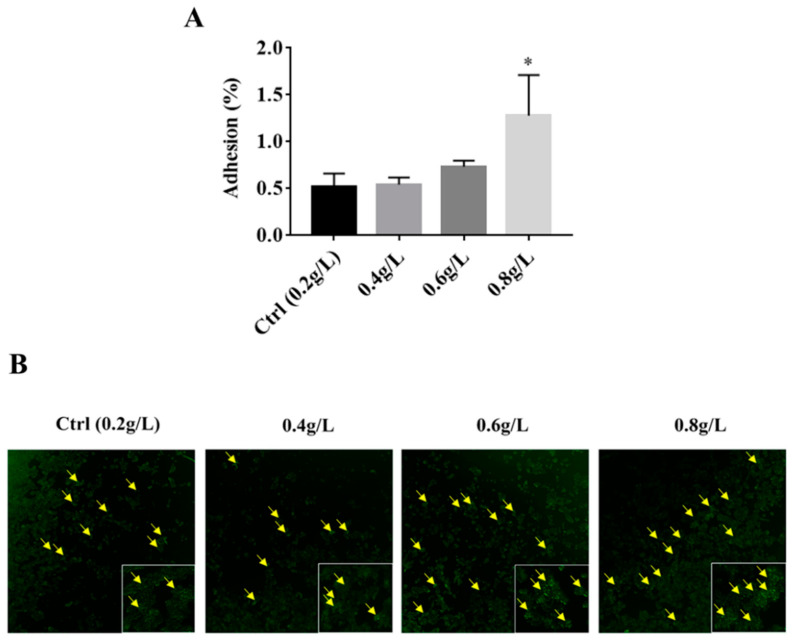
Effects of 0.2, 0.4, 0.6, and 0.8 g/L of tryptophan on adhesion of *A. muciniphila* BAA-835^T^. (**A**) Adhesion to HT-29, and (**B**) CFDA-SE-labeled *A. muciniphila* BAA-835^T^ adhere to HT-29. One-way ANOVA with a post hoc Dunnett test was used to determine statistical differences, compared with the Ctrl group. * *p* < 0.05.

**Figure 4 microorganisms-09-01511-f004:**
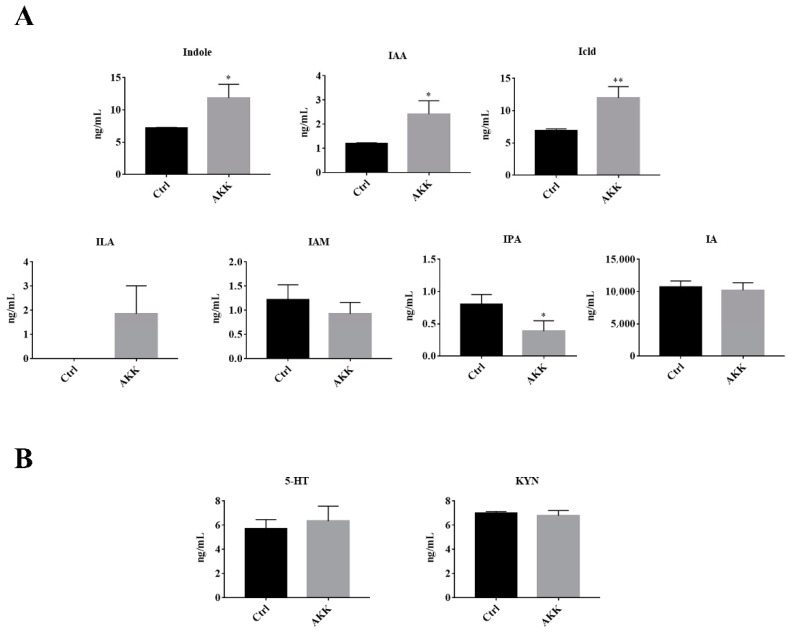
The concentrations of tryptophan metabolites in supernatant of *A. muciniphila* BAA-835^T^. (**A**) Indole and its derivatives. (**B**) 5-HT and KYN. Student’s *t*-test was used to determine statistical differences, compared with the Ctrl group. * *p* < 0.05, ** *p* < 0.01.

**Figure 5 microorganisms-09-01511-f005:**
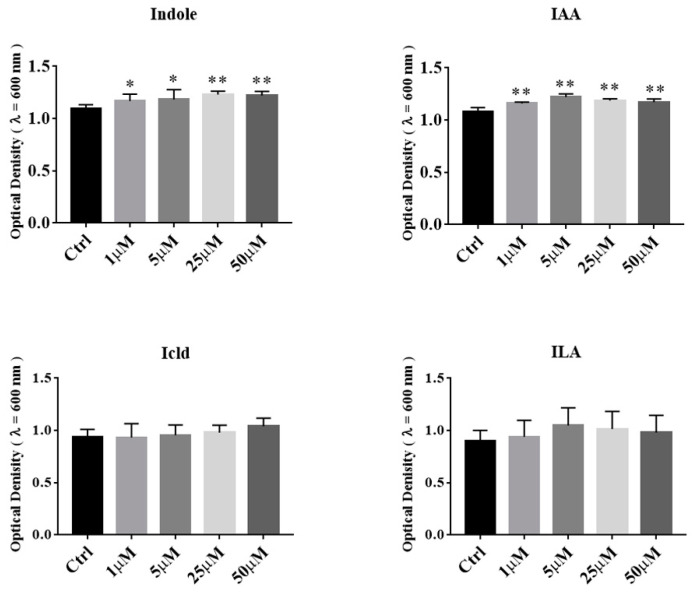
Effects of tryptophan metabolites produced by *A. muciniphila* on growth of *A. muciniphila* BAA-835^T^. One-way ANOVA with a post hoc Dunnett test was used to determine statistical differences, compared with the Ctrl group. * *p* < 0.05, ** *p* < 0.01.

**Figure 6 microorganisms-09-01511-f006:**
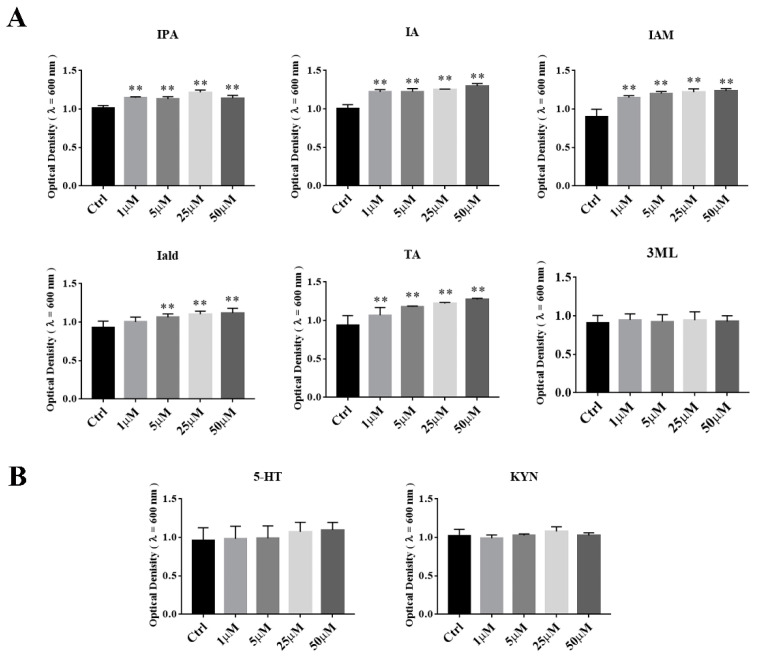
Effects of tryptophan metabolites produced by other gut microbiota on growth of *A. muciniphila* BAA-835^T^. (**A**) Indole and indole derivatives produced by gut microbiota. (**B**) 5-HT and KYN. One-way ANOVA with a post hoc Dunnett test was used to determine statistical differences, compared with the Ctrl group. ** *p* < 0.01.

## Data Availability

All data obtained in this study can be found in the manuscript or in the Appendix A.

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
