# Peer review of "Dose-Dependent Beneficial Effects of Tryptophan and Its Derived Metabolites on Akkermansia In Vitro: A Preliminary Prospective Study"

_microorganisms, 2021, doi:10.3390/microorganisms9071511_

Round 1

Reviewer 1 Report

This article "Dose-Dependent Beneficial Effects of Tryptophan and Its Derived Metabolites on Akkermansia in vitro: A Preliminary Prospective Study" investigated the effect of tryptophan on the growth of Akkermansia muchiniphila. This finding is expected to be a valuable source for the development of prebiotics. 

The topic is interesting and relevant, the manuscript is generally well-written. However, there are some points which should be improved prior to publication. I would recommend a more extensive description of the "3.2. The Effect of Tryptophan on the Autoaggregation, Hydrophobicity, and the Surface 164 Morphology of A. muciniphila" and "3.3. The Effect of Tryptophan on the Adhesion Ability of A. muciniphila to HT-29 Cells".

[3.2 section] Recent research (Becken, Bradford, et al. "Genotypic and Phenotypic Diversity among Human Isolates of Akkermansia muciniphila." Mbio 12.3 (2021): e00478-21.) found several phylogroup-specific phenotypes that may impact the colonization of the GI tract or modulate host functions, such as oxygen tolerance, adherence to epithelial cells, iron and sulfur metabolism, and bacterial aggregation based on genomic and phenotypic analysis.

I was wondering if autoaggregation and hydrophobicity experiments performed in this study can explain the colonization of the GI tract. The hydrophobicity test, which was performed in aqueous phase, was not rational to explain the colonization of the GI tract.

[3.3 section] The effect of supplementation of tryptophan on the adhesion of Akkermansia muciniphila to intestinal epithelial cells was described in the Figure 3. The adhesion ratio of Akkermansia muciniphila was significantly increased compared to the control group by additional 0.8 g/L tryptophan. Previous study (Reunanen, Justus, et al. "Akkermansia muciniphila adheres to enterocytes and strengthens the integrity of the epithelial cell layer." Applied and environmental microbiology 81.11 (2015): 3655-3662.) clearly showed the adhesion of Akkermansia muciniphila to the Caco-2 and HT-29 cell lines and to mucus. Lactobacillus rhamnosus GG was included in the experiments as a positive-control strain, since its ability to bind to human mucus and enterocytes has been well established. Notably, the Akkermansia muciniphila did not bind human colonic mucus.

Considering the previous study, I was wondering if the increased adhesion ability of Akkermansia muciniphila (from 0.5153% to 1.2789% in HT-29) by supplementation of tryptophan can lead to the meaningful promotion on colonization.

Author Response

Response to Reviewer 1 Comments

Point 1: I would recommend a more extensive description of the "3.2. The Effect of Tryptophan on the Autoaggregation, Hydrophobicity, and the Surface 164 Morphology of A. muciniphila" and "3.3. The Effect of Tryptophan on the Adhesion Ability of A. muciniphila to HT-29 Cells". 

Response 1: Thanks for your suggestion. We have added more description and made some modifications as suggested. (See Revised Manuscript, Page 4, Line 167-186; Page 5, Page 6.)

Point 2: [3.2 section] Recent research (Becken, Bradford, et al. "Genotypic and Phenotypic Diversity among Human Isolates of Akkermansia muciniphila." Mbio 12.3 (2021): e00478-21.) found several phylogroup-specific phenotypes that may impact the colonization of the GI tract or modulate host functions, such as oxygen tolerance, adherence to epithelial cells, iron and sulfur metabolism, and bacterial aggregation based on genomic and phenotypic analysis.

I was wondering if autoaggregation and hydrophobicity experiments performed in this study can explain the colonization of the GI tract. The hydrophobicity test, which was performed in aqueous phase, was not rational to explain the colonization of the GI tract.

Response 2: Thanks for your comments. Cell surface hydrophobicity and autoaggregation are important characteristics of potential probiotics as it indicates whether these strains can bind to the mucosal. It has been revealed that a large variety of surface glycoproteins are inserted into the hydrophobic cell wall of some microorganisms and they contribute to increase the likelihood of adhesion to cell receptors or proteins anchored in the cell wall. Meanwhile, improvement of autoaggregation ability can allow the colonisation and adherence of bacteria to epithelial cells, and resist the Infection by pathogenic bacteria pathogens. Thus, a preliminary analysis of autoaggregation and cell surface hydrophobicity of probiotics in vitro can assess the adhesion ability to GI. (das Neves Selis, N. et.al., Lactiplantibacillus plantarum strains isolated from spontaneously fermented cocoa exhibit potential probiotic properties against Gardnerella vaginalis and Neisseria gonorrhoeae. BMC Microbiol 2021, 21 (1), 198.) In this work, referring to methods of previous research, we tested the autoaggregation and cell surface hydrophobicity after supplementation of tryptophan in vitro (Braschi, G. et al., Effects of Sub-Lethal High Pressure Homogenization Treatment on the Adhesion Mechanisms and Stress Response Genes in Lactobacillus acidophilus 08. Front Microbiol 2021, 12, 651711.; Rungsirivanich, P. et al., Culturable Bacterial Community on Leaves of Assam Tea (Camellia sinensis var. assamica) in Thailand and Human Probiotic Potential of Isolated Bacillus spp. Microorganisms 2020, 8 (10).). The methods we used can be assumed that other factors have been ruled out, and only considered the impact of tryptophan on A. muciniphila. Further validation in vivo will be investigated in future. 

Point 3: [3.3 section] The effect of supplementation of tryptophan on the adhesion of Akkermansia muciniphila to intestinal epithelial cells was described in the Figure 3. The adhesion ratio of Akkermansia muciniphila was significantly increased compared to the control group by additional 0.8 g/L tryptophan. Previous study (Reunanen, Justus, et al. "Akkermansia muciniphila adheres to enterocytes and strengthens the integrity of the epithelial cell layer." Applied and environmental microbiology 81.11 (2015): 3655-3662.) clearly showed the adhesion of Akkermansia muciniphila to the Caco-2 and HT-29 cell lines and to mucus. Lactobacillus rhamnosus GG was included in the experiments as a positive-control strain, since its ability to bind to human mucus and enterocytes has been well established. Notably, the Akkermansia muciniphila did not bind human colonic mucus.

Considering the previous study, I was wondering if the increased adhesion ability of Akkermansia muciniphila (from 0.5153% to 1.2789% in HT-29) by supplementation of tryptophan can lead to the meaningful promotion on colonization.

Response 3: Thanks for your comments. Previous study showed that A. muciniphila has been found to inhabit the gastrointestinal (GI) tracts of more than 90% of adult subjects analyzed, and it constitutes 1 to 4% of the fecal microbiota (Collado, M. C. et.al., Intestinal integrity and Akkermansia muciniphila, a mucin-degrading member of the intestinal microbiota present in infants, adults, and the elderly. Appl Environ Microbiol 2007, 73 (23), 7767-70.). These data indicated that A.muciniphila can colonized in the intestine steadily. In healthy human colon, a thick mucus gel layer fully covers the epithelial cells, but in small intestine, the mucus layer is thinner and not continuous, which allows the bacteria contact with host enterocytes directly. Thus, colonizing bacteria can adhere either to the protective mucus gel covering the epithelial cell layer or directly to the enterocytes (Johansson, M. E. et al., Composition and functional role of the mucus layers in the intestine. Cell Mol Life Sci 2011, 68 (22), 3635-41). Reunanen, Justus, et al. found that A. muciniphila adheres strongly to the Caco-2 and HT-29 human colonic cell lines but not to human colonic mucus (Reunanen, Justus, et al. "Akkermansia muciniphila adheres to enterocytes and strengthens the integrity of the epithelial cell layer." Applied and environmental microbiology 81.11 (2015): 3655-3662). In this study, HT-29 cell lines were used to simulate intestinal epithelial cells in vivo, and the increased adhesion ability of Akkermansia muciniphila (from 0.5153% to 1.2789%) by supplementation of tryptophan was observed, these results suggested that A. muciniphila might bind to enterocytes directly in the small intestine. And the experiments in vivo will be done in future.

Reviewer 2 Report

The article entitled “Dose-Dependent Beneficial Effects of Tryptophan and Its Derived Metabolites on Akkermansia in vitro: A Preliminary Prospective Study” investigates the relationship between tryptophan and the probiotic Akkermansia muciniphila using in vitro culture methods. The authors found that the growth rate and the physiological functions of A. muciniphila were affected by tryptophan supplementation. These results are interesting as a preliminary study. On the other hand, I think the manuscript could be further improved by making some revisions.

Major issues

  1. Are the results of tryptophan supplementation to A. muciniphila specific? I think another microbe would need to be added or considered to show these results were A. muciniphila specific.
  2. To explain that the response of A. muciniphila is tryptophan specific, it would be better to consider the effect of other amino acids.

Minor issues

  1. In Materials and Methods, please check the 2.8. statistical analysis carefully. The authors seem to compare each group based on the control group in all multiple comparison. I thought the Tukey was not used as a post hoc test.
  2. In line 27, “the Verrucomicrobia phylum” should be corrected to “the phylum Verrcomicrobia”.
  3. In line 151, the authors wrote “As depicted in Figure 1A, there was no significant difference of OD600 between five groups in the stationary phase.”, but group number in Figure 1A was four. Please check and correct.
  4. In line 151, the authors wrote “there was no significant difference among groups in the stationary phase”. What definition is the stationary phase and what statistical method was used to compare?
  5. In Figre1B-C, dose dependency was not shown. The authors should discuss about it.
  6. In Figure 2, I think the statistical analysis about the cell size is necessary because morphological change by tryptophan supplementation is an important finding in this paper.
  7. In Discussion, the authors wrote that tryptophan could accelerate the rate of cell division, which leads to smaller cell morphology, and potentially improve the adhesion ability of A. muciniphila BAA-835T. But the tryptophan concentration in increase of A. muciniphila cell growth and the change of cell size was not the same.
  8. In Figure 3B, the authors should use the higher contrast images. I only recognized the yellow arrows.
  9. In all figure legend, the statistical analysis should be more described. It is not clear a certain group was significantly different compared with which group.

Author Response

Response to Reviewer 2 Comments

Point 1: Are the results of tryptophan supplementation to A. muciniphila specific? I think another microbe would need to be added or considered to show these results were A. muciniphila specific.

Response 1: Thanks for your comment. In our previous study, we observed that tryptophan supplementation diet could significantly enrich Akkermansia in vivo. Meanwhile, several studies showed that tryptophan metabolism was closely related to Akkermansia. Thus, this study aimed to investigate the reaction between tryptophan and A. muciniphila. Unfortunately, we have not considered that whether these results are A. muciniphila specific or not. According to the published literature, other probiotics, such as bifidobacterium, lactobacillus, and Faecalibacterium, can metabolize tryptophan to indoles, therefore, we speculated that there are reactions between tryptophan metabolism and these probiotics. We will explore these researches in future studies.

Point 2: To explain that the response of A. muciniphila is tryptophan specific, it would be better to consider the effect of other amino acids.

Response 2: Thanks for your comment. The growth, physiology and biochemistry of bacteria are always influenced by a variety of factors, such as pH, oxidative stress, bile salts, etc. However, in this work, we mainly aimed to explain the potential relationship between tryptophan metabolism and A. muciniphila, in order to propose a method for A. muciniphila to play a probiotic role in human health. There are a few researches about reaction between amino acid and probiotics, thus, we think it necessary to further investigate the effects of amino acids on growth, physical and biochemical of probiotics.

Point 3: In Materials and Methods, please check the 2.8. statistical analysis carefully. The authors seem to compare each group based on the control group in all multiple comparison. I thought the Tukey was not used as a post hoc test.

Response 3: Thanks for your comment. Results contains multiple groups were compared with the one-way ANOVA test followed by the Tukey post hoc test. But we think it appropriately to use one-way ANOVA test followed by the Dunnett post hoc test as you suggested. Thus, we reanalyzed the data in section 3.5 (Figure 5) and 3.6 (Figure 6). The figures were readjustment, and the significance was remarked. (See Revised Manuscript, Section 3.5 and 3.6, Page 7-9).

Point 4: In line 27, “the Verrucomicrobia phylum” should be corrected to “the phylum Verrcomicrobia”.

Response 4: Thanks for your comment. We have made some modifications as suggested. (See Revised Manuscript, Page 1, Line 30). 

Point 5: In line 151, the authors wrote “As depicted in Figure 1A, there was no significant difference of OD600 between five groups in the stationary phase.”, but group number in Figure 1A was four. Please check and correct.

Response 5: Thanks for your careful comment. We have checked and made some modifications as suggested. (See Revised Manuscript, Page 4, Line 154).

Point 6: In line 151, the authors wrote “there was no significant difference among groups in the stationary phase”. What definition is the stationary phase and what statistical method was used to compare?

Response 6: Thanks for your comment. Generally, microbial growth curve contains four stages, lag phase, log phase, stationary phase, and declining phase. In this study, we tested OD600 in stationary phase, and analyzed the data used one-way ANOVA followed by Dunnett pot hoc test. However, there was no significant difference compared with Ctrl group in stationary phase.

Point 7: In Figre1B-C, dose dependency was not shown. The authors should discuss about it.

Response 7: Thanks for your comment. In our previous study, we investigated effects of supplementation of tryptophan to D-galactose induced aging mice, and we found that 4g/kg tryptophan diet (add 2 g/kg tryptophan to Ctrl diet) could improve the relative abundance of Akkermansia in aging mice, but not 8 g/kg tryptophan diet group (Yin, J. et al., Ameliorative Effect of Dietary Tryptophan on Neurodegeneration and Inflammation in d-Galactose-Induced Aging Mice with the Potential Mechanism Relying on AMPK/SIRT1/PGC-1alpha Pathway and Gut Microbiota. J Agric Food Chem 2021, 69 (16), 4732-4744.). In this study, our results indicated that 4 g/L tryptophan in BHI medium could significantly promote the growth of Akkermansia muciniphila BAA-835T, this was consistent with the previous study in vivo. We have made some modifications as suggested. (See Revised Manuscript, Page 9, Line 281-286).

Point 8: In Figure 2, I think the statistical analysis about the cell size is necessary because morphological change by tryptophan supplementation is an important finding in this paper.

Response 8: Thanks for your comment. We analyzed the cell size in SEM pictures by Nano Measurer software, and the statistical analysis was performed by one-way ANOVA, following by Dunnett pot hoc test. Meanwhile, we have made some modifications as suggested. (See Revised Manuscript, Page 4-5, Line 183-191, Figure 2).

Point 9: In Discussion, the authors wrote that tryptophan could accelerate the rate of cell division, which leads to smaller cell morphology, and potentially improve the adhesion ability of A. muciniphila BAA-835T. But the tryptophan concentration in increase of A. muciniphila cell growth and the change of cell size was not the same.

Response 9: Thanks for your comment. In this study, we test the adhesion ability and growth curve for three times, and observed that 0.4g/L tryptophan could promote the growth in 24h. However, we regrettably found that 0.4 g/L tryptophan could not shorten the cell morphology, but 0.6 g/L and 0.8 g/L tryptophan group showed a smaller cell morphology, and a higher adhesion ability. Previous study observed that supplementation of mucin could shorten the size of A. muciniphila, and the author speculated that smaller diameter possibly due to faster the cell division (Liu, X. et al., Transcriptomics and metabolomics reveal the adaption of Akkermansia muciniphila to high mucin by regulating energy homeostasis. Sci Rep 2021, 11 (1), 9073.). However, we found that 0.4g/L tryptophan could promote the growth of A. muciniphila, but the cells in this group owned a bigger size, compared with other groups. Significantly, Schaechter, M. et al., proposed that the faster a bacterial cell grows, the bigger the cell morphology, and comply with SMK growth law (Schaechter, M. et al., Dependency on medium and temperature of cell size and chemical composition during balanced grown of Salmonella typhimurium. J. Gen. Microbiol. 19, 592–606 (1958)), although SMK has been corrected by Hai Zheng et al. (Zheng, H. et al., General quantitative relations linking cell growth and the cell cycle in Escherichia coli. Nat Microbiol 2020, 5 (8), 995-1001). These investigations support our results effectively. Thus, views on the relationship between cell size and growth were not fully consistent and need to do further research. Our results suggested that tryptophan has multiple effects on the growth and adhesion of Akkermansia muciniphila, the transcriptional level should be further investigated by RNA-Seq in future. Meanwhile, we made some modifications in discussion section as you suggested. (See Revised Manuscript, Page 10, Line 298-308).

Point 10: In Figure 3B, the authors should use the higher contrast images. I only recognized the yellow arrows.

Response 10: Thanks for your comment. The figure was readjustment in revised manuscript. (See Revised Manuscript, Page 6, Figure 3).

Point 11: In all figure legend, the statistical analysis should be more described. It is not clear a certain group was significantly different compared with which group.

Response 11: Thanks for your comment. We have made some modifications as suggested. (See Revised Manuscript, Figure captions).

Reviewer 3 Report

In this preliminary study, the authors explore the effects of tryptophan and its derived metabolites on Akkermansia muciniphila, as potential probiotic bacteria species. The methods or assays were well selected to evaluate this aim. Overall, the manuscript was well-drafted and described. However, some substantial revisions need to be done to improve the quality.

1) Only one strain of A. muciniphila (BAA-835T) and one epithelial cell line (HT-29) were utilized to explore the potential effect of tryptophan. In addition, in some assays such as in Figure 1, only triplicate samples were tested as described in the methods without any repeat. Some results showed not consistent. Figure 1 data showed that 0.4 g/L of Trp treatment significantly increased the proliferation of A. muciniphila. However, the SEM figures shown in Figure 2C indicated that 0.6 g/L of Trp treatment induced more dividing cells of A. muciniphila, as described by the authors with smaller diameters.

2) The quality of Figure 3B need to be improved. Fluorescence-labeling bacteria show too dim color, and the cultured HT-29 cells were not distributed equally in the 24-well, resulting in the attached bacteria either located in the center or only on side of the well. In addition, some arrows (legend should include the meaning of arrows) pointed outside of the pictures. The HT-29 cells are better stained with orange cell Tracker, which showed red color under fluorescence microscopy to well differentiate with bacterial stain.

3) Figure 4 was divided into Figure 4A and 4B. However, in the context, no difference was mentioned.

4) In section 3.5., the authors described the tryptophan metabolites produced by Akkermansia on the growth of A. municiphila BAA-835T. However, the metabolites indole, IAA, etc. were directly added in the BHI medium. No description was made that they are produced by Akkermansia. Similarly, in section 3.6., the authors described the additive metabolites were produced by other gut microbiota, which was not correct.

5) There are two IAs in line 247, one should be TA?

6) In line 324, L should be capitalized in Listeria

7) In the conclusion, the authors concluded that supplementation of tryptophan or its metabolites may promote the beneficial bacteria and inhibit the intestinal adhere of pathogenic bacteria, without any data support or discussion.

Author Response

Response to Reviewer 3 Comments

Point 1: Only one strain of A. muciniphila (BAA-835T) and one epithelial cell line (HT-29) were utilized to explore the potential effect of tryptophan. In addition, in some assays such as in Figure 1, only triplicate samples were tested as described in the methods without any repeat. Some results showed not consistent. Figure 1 data showed that 0.4 g/L of Trp treatment significantly increased the proliferation of A. muciniphila. However, the SEM figures shown in Figure 2C indicated that 0.6 g/L of Trp treatment induced more dividing cells of A. muciniphila, as described by the authors with smaller diameters.

Response 1: Thanks for your comment. In this study, we test the adhesion ability and growth curve for three times, and observed that 0.4 g/L tryptophan could promote the growth in 24h. However, we regrettably found that 0.4 g/L tryptophan could not shorten the cell morphology, but 0.6 g/L and 0.8 g/L tryptophan group showed a smaller cell morphology, and a higher adhesion ability. Previous study observed that supplementation of mucin could shorten the size of A. muciniphila, and the author speculated that smaller diameter possibly due to faster the cell division (Liu, X. et al., Transcriptomics and metabolomics reveal the adaption of Akkermansia muciniphila to high mucin by regulating energy homeostasis. Sci Rep 2021, 11 (1), 9073.). However, we found that 0.4g/L tryptophan could promote the growth of A. muciniphila, but the cells in this group owned a bigger size, compared with other groups. Significantly, Schaechter, M. et al., proposed that the faster a bacterial cell grows, the bigger the cell morphology, and comply with SMK growth law (Schaechter, M. et al., Dependency on medium and temperature of cell size and chemical composition during balanced grown of Salmonella typhimurium. J. Gen. Microbiol. 19, 592–606 (1958)), although SMK has been corrected by Hai Zheng et al. (Zheng, H. et al., General quantitative relations linking cell growth and the cell cycle in Escherichia coli. Nat Microbiol 2020, 5 (8), 995-1001). These investigations support our results effectively. Thus, views on the relationship between cell size and growth were not fully consistent and needs to do further research. Our results suggested that tryptophan has multiple effects on the growth and adhesion of Akkermansia muciniphila, the transcriptional level should be further investigated by RNA-Seq in future. Meanwhile, we made some modifications in discussion section as you suggested. (See Revised Manuscript, Page 10, Line 298-308).

Point 2: The quality of Figure 3B need to be improved. Fluorescence-labeling bacteria show too dim color, and the cultured HT-29 cells were not distributed equally in the 24-well, resulting in the attached bacteria either located in the center or only on side of the well. In addition, some arrows (legend should include the meaning of arrows) pointed outside of the pictures. The HT-29 cells are better stained with orange cell Tracker, which showed red color under fluorescence microscopy to well differentiate with bacterial stain.

Response 2: Thanks for your careful comments. The figure was readjustment in revised manuscript. According to previous study, bacteria can be labelled by CFDA-SE, so we choose this fluorochrome (Wang, G.et al., Colonisation with endogenous Lactobacillus reuteri R28 and exogenous Lactobacillus plantarum AR17-1 and the effects on intestinal inflammation in mice. Food Funct 2021, 12 (6), 2481-2488.). And as you suggested, green color was not easy to photograph by microscopy, the red/blue color will be first considered in future work.

Point 3: Figure 4 was divided into Figure 4A and 4B. However, in the context, no difference was mentioned.

Response 3: Thanks for your careful comment. We have made some modifications as suggested. (See Revised Manuscript, Page 6-7, Line 227-236).

Point 4: In section 3.5., the authors described the tryptophan metabolites produced by Akkermansia on the growth of A. municiphila BAA-835T. However, the metabolites indole, IAA, etc. were directly added in the BHI medium. No description was made that they are produced by Akkermansia. Similarly, in section 3.6., the authors described the additive metabolites were produced by other gut microbiota, which was not correct.

Response 4: Thanks for your comment. In section 3.4, the HPLC/Q-TRAP MS method was used to analyze the contents of 9 common tryptophan metabolites produced by Akkermansia muciniphila BAA-835T, we found that A. muciniphila can utilize the tryptophan to produce indole, IAA, Icld, and ILA. In order to further investigate the reaction between A. muciniphila and tryptophan metabolites, we added the indole, IAA, Icld, and ILA into BHI, and cultured with A. muciniphila BAA-835T, our results showed that indole and IAA could significantly promote the growth of A. muciniphila BAA-835T. According to the fact that numerous of microorganisms colonize the human gastrointestinal tract, previous study has proved that tryptophan can be metabolized to indoles and its derivatives by gut microbiota, such as bifidobacterium, lactobacillus, and Faecalibacterium (Roager, H. M. et al., Microbial tryptophan catabolites in health and disease. Nat Commun 2018, 9 (1), 3294.). Thus, we also added indoles produced by other bacteria to BHI medium to investigate the effects of these metabolites on growth of A. muciniphila (section 3.6).

Point 5: There are two IAs in line 247, one should be TA?

Response 5: Thanks for your careful comment. We change one of IA to TA as suggested. (See Revised Manuscript, Page 8, Line 264).

Point 6: In line 324, L should be capitalized in Listeria.

Response 6: Thanks for your careful comment. We have made some modifications as suggested. (See Revised Manuscript, Page 10, Line 348).

Point 7: In the conclusion, the authors concluded that supplementation of tryptophan or its metabolites may promote the beneficial bacteria and inhibit the intestinal adhere of pathogenic bacteria, without any data support or discussion.

Response 7: Thanks for your comment. Our expression is not exact and accurate, we have made some modifications as suggested. (See Revised Manuscript, Page 11, Line 392-395).

Round 2

Reviewer 2 Report

Thank you for responding my comment. The manuscript is well written and I have no comments.